# Connexin46 Expression Enhances Cancer Stem Cell and Epithelial-to-Mesenchymal Transition Characteristics of Human Breast Cancer MCF-7 Cells

**DOI:** 10.3390/ijms222212604

**Published:** 2021-11-22

**Authors:** Rodrigo A. Acuña, Manuel Varas-Godoy, Diego Herrera-Sepulveda, Mauricio A. Retamal

**Affiliations:** 1Centro de Medicina Regenerativa, Facultad de Medicina, Clínica Alemana Universidad del Desarrollo, Santiago 7780272, Chile; 2Centro de Biología Celular y Biomedicina (CEBICEM), Facultad de Medicina y Ciencia, Universidad San Sebastian, Santiago 7510157, Chile; manuel.varas@uss.cl; 3Carrera de Medicina Universidad del Desarrollo, Facultad de Medicina, Clínica Alemana Universidad del Desarrollo, Santiago 7780272, Chile; dherreras@udd.cl; 4Centro de Fisiología Celular e Integrativa, Facultad de Medicina Clínica Alemana, Universidad del Desarrollo, Santiago 7780272, Chile; 5Programa de Comunicación Celular en Cáncer, Facultad de Medicina Clínica Alemana, Universidad del Desarrollo, Santiago 7780272, Chile

**Keywords:** Connexin46, breast cancer, cancer stem cells, epithelial-to-mesenchymal transition, stemness

## Abstract

Connexins (Cxs) are a family of proteins that form two different types of ion channels: hemichannels and gap junction channels. These channels participate in cellular communication, enabling them to share information and act as a synchronized syncytium. This cellular communication has been considered a strong tumor suppressor, but it is now recognized that some type of Cxs can be pro-tumorigenic. For example, Cx46 expression is increased in human breast cancer samples and correlates with cancer stem cell (CSC) characteristics in human glioma. Thus, we explored whether Cx46 and glioma cells, can set up CSC and epithelial-to-mesenchymal transition (EMT) properties in a breast cancer cell line. To this end, we transfected MCF-7 cells with Cx46 attached to a green fluorescent protein (Cx46GFP), and we determined how its expression orchestrates both the gene-expression and functional changes associated with CSC and EMT. We observed that Cx46GFP increased Sox2, Nanog, and OCT4 mRNA levels associated with a high capacity to form monoclonal colonies and tumorspheres. Similarly, Cx46GFP increased the mRNA levels of n-cadherin, Vimentin, Snail and Zeb1 to a higher migratory and invasive capacity. Furthermore, Cx46GFP transfected in MCF-7 cells induced the release of higher amounts of VEGF, which promoted angiogenesis in HUVEC cells. We demonstrated for the first time that Cx46 modulates CSC and EMT properties in breast cancer cells and thus could be relevant in the design of future cancer therapies.

## 1. Introduction

Connexins (Cxs) are a family of transmembrane proteins that have the unique property to form two different types of ion channels: hemichannels and gap junction channels [1]. Hemichannels are formed by six Cx subunits and, when they open, communicate the cytoplasm with the extracellular milieu [2]. When two hemichannels coming from two different cells make contact, they form a gap junction channel, and due to its dispositions at the plasma membrane, these channels allow the exchange of ions and molecules up to ~1.5 kDa between the cytoplasms of these neighboring cells [1]. Under normal conditions, almost all cell types in the human body express at least one type of Cx and therefore have the capability to interconnect with other cells, share information and act as a synchronized syncytium. This cellular communication has been considered as a tumor suppressor [3], and a decrease in Cx expression has been strongly correlated with the appearance of several types of cancer [4]. However, currently it is recognized that there are some types of Cxs that seem to be pro-tumorigenic [4]; one of these, under physiological conditions, Cx46 is almost exclusively expressed in the eye lens [5]. However, this Cx may also have an important role in cancer progression. Thus, despite the seminal studies using animal models of lung and bone cancer in which a decrease of Cx46 mRNA levels was observed [6,7], more recent studies show that Cx46 protein levels are increased in human samples of infiltrating breast carcinoma [8] and glioblastoma [9]. In human glioblastoma, Cx46 expression was correlated with cancer stem cell (CSC) proliferation, self-renewal, and propagation, and negatively correlated with patient survival [9,10]. Similarly, the use of siRNA against Cx46 inhibited the tumor growth of MCF-7 (breast adenocarcinoma cell line) [8] and Y79 (human retinoblastoma cell line) [11] cells in a mouse xenograft model. Because it is very well known that only CSCs can initiate a tumor in xenograft models [12,13,14], the presence of Cx46 in human cancer could be considered a factor that increases CSC phenotypes in certain types of cancer such as breast cancer. Additionally, to metastasize, cancer cells must gain migratory and invasive characteristics, and they can do that through a process known as the epithelial-to-mesenchymal transition (EMT) [15]. Cxs have been reported to participate in EMT in cancer related processes [16]. Despite this evidence, as far as we know the effect of Cx46 at the cellular and molecular levels in breast cancer cells has never been explored, nor has its relationship with the properties of CSC and EMT been established. Therefore, we explored how the expression of Cx46 in a human breast cancer cell line could enhance its CSC and EMT properties. To this end, we transfected MCF-7 cells with Cx46 attached to the green fluorescent protein MCF-7Cx46GFP, and we determined how its expression orchestrates both gene-expression and functional changes associated with CSC and EMT. We observed that Cx46GFP expression induced an increase of mRNA levels of CSC transcription factors such Sox2, Nanog, and OCT4, which were associated with a high capacity to form monoclonal colonies and tumorspheres. On the other hand, Cx46GFP increased the mRNA levels of n-cadherin, Vimentin, Snail, and Zeb1, according a higher migratory and invasive capacity. Interestingly, the supernatant of MCF-7Cx46-GFP presents higher amounts of VEGF, which was associated with an angiogenic effect when it was placed over HUVEC cells. For the first time, we showed a potent effect of Cx46 on a cell line derived from breast cancer in terms of CSC and EMT induction, and these results are congruent with previous results obtained in glioma. Thus, our work supports the hypothesis that Cx46 is pro-tumorigenic in diverse types of cancers and should be considered in the design of future cancer therapies.

## 2. Results

### 2.1. Over-Expression of Cx46 Does Not Modify the Rate of MCF-7 Cell Division

Recently, Cx46 has been involved in the maintenance of CSCs in glioblastoma [9]; however, the effect of Cx46 on breast cancer stem cell properties was unknown. To explore this possibility, we transfected Cx46GFP in MCF-7 cells, which is a cell line derived from a hormone dependent ductal adenocarcinoma [17]. Under control conditions MCF-7 cells express very low levels of Cx46, which, in Western blot (WB) analyses, appears to have ~50 kDa (Figure 1A). As expected, MCF-7 cells transfected with Cx46GFP (MCF-7Cx46-GFP) present a prominent band of ~68 kDa, which is the sum of the molecular weights of Cx46 and GFP (Figure 1A). Also, a band of about 50 kDa was observed in these cells, most likely corresponding to the Cx46 endogenously expressed and/or due to a proteolytic cleavage of GFP, which has been observed in other systems [18,19]. As a comparative control of aggressiveness, we used a triple negative cell line known as MDA-MB-231 [20,21]. This cell line does not express Cx46. However, the Cx43 expression was observed (Figure 1A). Because one of the major cancer cell features is their high rate of proliferation, we tested whether Cx46 can enhance this property in MCF-7 cells. Under our control conditions, the MCF-7, MCF-7Cx46-GFP, and MDA-MB-231 cell lines showed the same rate of cell division after 5 days in culture (Figure 1B). Thus, Cx46 expression was not associated with increased or varied changes in cell division. MCF-7Cx46-GFP observed under a fluorescent microscope showed that Cx46 was located mostly in a cytoplasmic disposition (Figure 1C), accordingly gap junction-like structures were not often observed.

### 2.2. Cx46 Increases the Expression of CSC Related Genes and Funtional Characteristics

It is well known that CSC stemness has been related to poor patient prognosis in several types of cancers [22,23,24] whose properties are driven by transcription factors, the most important being OCT4, Sox2, and Nanog [25]. In our breast cancer cell model, the expression of Cx46GFP increased mRNA levels of OCT4 and Sox2; however, the Nanog mRNA levels were unchanged (Figure 2A). The analysis of these results indicated that OCT4 and Sox2 increased 3.7 ± 0.8 and 2.4 ± 0.4 times their levels, respectively, compared to non-transfected MCF-7 cells. Both OCT4 and Sox2 have been strongly correlated to the acquisition of CSC characteristics in breast cancer cells [26,27]. If Cx46 increases these CSC-associated transcription factors, it could be expected that the MCF-7 expression of this protein would show enhanced CSC functional characteristics, such as formation of clonogenic colonies and tumorspheres [28,29]. The clonal growth of MCF-7 was 40% lower compared to MCF-7Cx46-GFP capacity (Figure 2B,C). Additionally, MCF-7 and MCF-7Cx46-GFP were cultured in ultralow attachment plates with tumorsphere supplements for two weeks, and under this condition Cx46-expressing cells formed 36% larger tumorspheres compared to those from MCF-7 (WT) (Figure 2D,E). These results confirm that Cx46 expression is sufficient to induce a CSC-like phenotype in hormone-dependent breast cancer cells.

### 2.3. Cx46 Increases the Expression of EMT Related Genes and Functional Characteristics

Another important aspect of cancer progression is the property of some cells to detach from the primary tumor and migrate to a local lymphovascular system, and then to distant organs in a process known as metastasis [30]. This process is so important that most of deaths associated with cancer results from it [30]. Therefore, we explored the effect of Cx46 expression on the EMT process in MCF-7 WT cells and found that the expression of Cx46GFP increased the mRNA levels of Snail (1.7 ± 0.4 times), Zebt1 (1.9 ± 0.2 times), n-cadherin (3.8 ± 1.0 times), and vimentin (15.1 ± 3.0 times), all of which are well known EMT-related genes (Figure 3A). The capacity to migrate and invade new tissues is characteristic of mesenchymal-like cells [31,32]. We then compared the MCF-7 WT, MCF-7Cx46-GFP, and MDA-MB-231 cell capacity to migrate and invade in vitro. To explore migration, we performed a scratch assay [33]. We studied the migratory capacity of these three breast cancer cell lines up to 16 hrs. After this time, it was evident that MCF-7Cx46-GFP migrated faster than MCF-7 WT cells (Figure 3B). Interestingly, the rate of migration of MCF-7Cx46-GFP was not different compared to highly migrative MDA-MB-231 (Figure 3C). However, metastatic cells not only need to move faster but also need to release metalloproteinases to pass between the tissues and thus be able to reach the lymphatic and/or blood tissue. To study cancer cell invasive capacity, cells were placed in an extracellular-like matrix and were induced to actively pass through it following attractants [34]. Our results show that 22 ± 2 of MCF-7 WT cells were able to cross the Matrigel after 24 h (Figure 3D,E). In the same time frame, 58 ± 6 MCF-7Cx46-GFP and 115 ± 10 MDA-MB-231 cells could also cross (Figure 3D,E). Because is known that transfection *per se*, can affect some cell properties, we performed transfection control using plasmids coding for GFP alone (MCF-7 GFP). Additionally, if our results were indeed because of Cx46, we transfected MCF-7 with Cx46 without the GFP tag (as a positive control of the MCF-7Cx46-GFP results). Our transfection with GFP resulted in cells expressing different degrees of GFP (Appendix A). Also, Western blot analyses revealed that MCF-7 transfected with Cx46 expressed large amounts of this protein, represented as a prominent band in 46 kDa in each of the three experiments performed. Several bands with lower molecular weights were also observed. As expected, neither MCF-7 or MCF-7 transfected with GFP showed detectable expression of Cx46. (Appendix A). In the migration experiments, MCF-7Cx46 show a faster migration velocity than the non-transfected MCF-7 or MCF-7 transfected with GFP alone (Appendix A), strongly supporting our results obtained using MCF-7Cx46-GFP. The analyzes of these results revealed that MCF-7 and MCF-7 GFP after 16 h closed ~20% of the initial wound area. However, MCF-7 Cx46 closed ~80% of the initial area in the same period of time (Appendix A). Similar results were obtained using MCF-7Cx46-GFP cells (see above). These results suggest that Cx46 increases the EMT characteristics of breast cancer cells and therefore could be associated with metastasis in hormone dependent breast cancer. 

### 2.4. Cx46 Increases the Release of VEGF and Promote Tube Formation in Endothelial Cells

Angiogenesis is a very important process that enables tumor growth [35], and the production of angiogenic molecules has been associated with CSC niches [35]. Because MCF-7Cx46-GFP presents a higher CSC stemness than its WT counterpart (Figure 2), we explored whether these cells could release angiogenic factors involved in both processes, such as the master regulator of angiogeneisis VEGF. In order to study whether MCF-7Cx46-GFPs are producing VEGF, the expression of VEGF in whole cell extracts was determined by Western blot. As expected, an increase of three times the content of VEGF in MCF-7Cx46-GFP cells compared to the MCF-7 WT cells (Figure 4A) was observed. To corroborate that these cells are not only producing but also releasing VEGF, we measured their levels in MCF-7Cx46-GFP and MCF-7 WT in the supernatant. We observed that MCF-7Cx46-GFP cells released almost twice the VEGF compared to MCF-7 WT cells (200 ± 20 v/s 110 ± 20 pg/mL respectively) (Figure 4B). Additionally, we explored whether the VEGF released was able to induce angiogenic-like modifications in HUVEC cells. After 6 h post exposure to the MCF-7 WT- and MCF-7Cx46-GFP-conditioned media, HUVEC cells present a significant increase in angiogenic modifications, including more and larger junctions (70 ± 7% increase), nodes (34 ± 4% increase), meshes (216 ± 10% increase), segments (104 ± 4% increase), and branches (54 ± 19% increase) compared with the conditioned media obtained from MCF-7 WT cells (Figure 4C,D). All these results support a pro-angiogenic role of Cx46 when expressed in hormone-dependent breast cancer cells.

## 3. Discussion

In this work we tested the hypothesis that overexpression of Cx46 can increase the CSC, EMT, and angiogenic characteristics of a human breast cancer cell line known as MCF-7 cells. We observed that the expression of Cx46GFP or Cx46 alone increased CSC and EMT-associated transcription factors, in addition to increasing the expression and release of VEGF angiogenic factor. In agreement with this, MCF-7Cx46-GFP cells presented functional characteristics associated to EMT, CSC, and angiogenic processes. For example, they formed larger tumor spheres and more monoclonal colonies, and they migrated and invaded more than the non-transfected MCF-7 cells. Also, MCF-7Cx46-GFP conditioned media was able to induce angiogenic-like modifications in HUVEC cells, unlike the MCF-7 non-transfected conditioned media.

CSCs are distinguished for being a small population inside of a tumor, but are very relevant for patient survival, since CSCs are directly responsible for tumor growth, chemoresistance, metastasis, and relapse [22,23,24,36,37]. Accordingly, CSCs have become important targets for new cancer therapies [38,39,40]. Cx46 is very well correlated with CSC properties in human glioma [9,10]. Last year, it was suggested that Cx46 gap junction channels mediate this phenomenon; however, the mechanism behind this cellular transformation was not studied [10]. One possibility is that Cx46 can interact with Sox2, Nanog, or OCT4, increasing their role as transcription factors in proteins needed for CSC phenotype acquisition. Thus, for example, Cx26 in triple negative breast cancer cells promotes CSC self-renewal by forming a complex with Nanog and focal adhesion kinase, resulting in Nanog stabilization [41]. Cx46 has been localized in the nucleus of anterior pituitary folliculostellate cells, where it co-localized with Nopp-140, a nucleolar factor involved in rRNA processing [42]. Thus, nuclear localization of Cx46 and interaction with Sox2, Nanog, or OCT4 can potentially explain our results.

Another important aspect of cancer is the property of some cells to detach from the primary tumor and migrate to the local lymphovascular system, and then to distant organs in a process known as metastasis [30]. To have the ability to migrate, some cells have to lose their polar epithelial phenotype and gain a migratory mesenchymal phenotype in a process known as EMT [30]. Different transcription factors have been suggested to orchestrate the gene expression change associated with EMT, among which, Snail [43,44,45,46], Twist [46,47,48,49], and ZEB1 [46,50,51] have critical roles. It has been shown that Cx26 upregulates the signaling pathways associated with PI3K/AKT, which in turn increases the EMT characteristics of non-small-cell lung cancer [52]. In the case of Cx46, we recently demonstrated that MCF-7Cx46-GFP cells release extracellular vesicles (EVs, mostly exosomes) containing Cx46 in their membranes [53]. Exosomes contain different cargo molecules, among these, MicroRNAs (miRNAs) [54,55,56,57]. The presence of miRNAs has taken on an important role due to their ability to regulate gene expression post-transcriptionally. The presence of the enhanced transfer by Cx46 of information from EVs to recipient cells was associated with an increase of the migratory and invasive capacities of the MCF-7 WT recipient cells [53]. Thus, one possibility to explain our results is a positive loop in which Cx46-positive cells release the EMT-associated miRNAs contained in EVs, which transform or enhance EMT characteristics in Cx46-negative cells.

Angiogenesis is a key process in breast cancer growth, invasion, and metastasis [58]. In breast cancer, mostly the role of Cx43 has been studied. In this case, downregulation of Cx43 in human breast cancer cell lines (MDA-MB-231 and Hs578T) was correlated with an increase of thrombospondin-1 and VEGF, two antiangiogenic molecules [59]. Moreover, downregulation of Cx43 in endothelial cells induced by MDA-MB-231 conditioned media induced an increase of endothelial cells proliferation [60]. On the other hand, Cx43 has been identified for its role in reducing the expression of hypoxia induced factor (HIF-1α) by the downregulation of VEGF in murine cells [61]. 

In our case, an increase of Cx46 was associated with a higher release of VEGF and an induction of angiogenesis in HUVEC cells. Cx46 was originally detected in lens cells, which is an avascular tissue [62] and implicated in early breast tumor growth [63].

Interestingly, renal cancer stem cells release EVs that induce angiogenesis [64,65]. It appears highly likely that the MCF-7Cx46-GFP released EVs [53] which could contain information such as miRNAs that induce VEGF, and other factors associated to angiogenesis, such as HIF-1α.

Thus, this research is the first to demonstrate that Cx46 expression can be associated with a growth of vasculature. We suggest that the same molecular mechanisms that operate in CSC and EMT increase the transcription factors induced by Cx46, which can also function as a pro-angiogenic factor.

## 4. Materials and Methods

### 4.1. Plasmid and Cells

Stable transfection was performed used a pCMV6-AC-Cx46GFP cloning vector from OriGene Technologies (Origene, Rockville, MD, USA). MCF-7 cells (ATCC) were transfected with Lipofectamine 2000 (Life Technologies, Carlsbad, CA, USA), according to the manufacturer’s instructions. Transfected cells were selected with 300 μg/mL of G418 (Gibco). Stable colonies were isolated two weeks later and propagated in DMEM containing G418 at 50 μg/mL. The resulting clone was named MCF-7Cx46-GFP. Transient transfection was performed used a human Cx46 in pSFFV-neo mammalian expression plasmid (addgene, Watertown, MA, USA), kindly provided by Dr. Viviana Berthoud (Chicago University, Pediatrics-Hematology and Oncology Departments, Chicago IL). Breast cancer cells lines were purchased from the American Type Culture Collection (ATCC). MCF-7, MCF-7Cx46-GFP and MDA-MB-231 were maintained in Dulbecco’s modified Eagle’s medium (DMEM) (Life Technologies, Carlsbad, CA, USA), supplemented with 10% heat inactivated fetal bovine serum (FBS) (Life Technologies, Carlsbad, CA, USA) and penicillin/Streptomycin (100 U/mL, 100 μleme Streptomycin). Cells were maintained at 37 °C in a 5% CO_2_ humidified incubator.

### 4.2. Western Blot

Briefly, cells were lysed in RIPA buffer supplemented with protease inhibitors (Roche). The protein concentration was determined using a protein assay kit (ThermoFisher Scientific, Waltham, MA, USA) and read in a Qubit 3.0 Fluorometer (ThermoFisher Scientific, Waltham, MA, USA). Of the total protein from the cell lysates, 100 μg was resolved with 10% SDS PAGE gel by PAGE and transferred to a nitrocellulose membrane (Bio-Rad, Hercules, CA, USA). Membranes were incubated with the following primary antibodies: anti Cx46 (Santa Cruz Biotechnology; 1:500), anti Cx43 (Santa Cruz Biotechnology; 1:500), and beta actin (Santa Cruz Biotechnology, 1:5000). All secondary antibodies were horseradish protein (HRP) conjugates (Abcam). Protein bands were detected using Immobilon Forte Western HRP substrate (Millipore, Burlington, MA, USA) and visualized with LI-COR C-Digit Chemiluminescense Western Blot Scanner systems (LI-COR, Inc., Lincoln, NE, USA).

### 4.3. Clonogenic Assay

The MCF-7 and MCF-7Cx46-GFP breast cancer cell lines were seeded in 35 mm culture plates, 200 cells were seeded in triplicate. The cells were kept at 37 °C in 5% CO_2_ for 14 days, and the culture media were changed every 2 days. After 14 days, the cells were washed with 1X PBS, fixed with 70% ethanol, and stained with 0.5% crystal violet. The colonies contained in each plate were photographed using a magnifying glass, to later be incubated with lysis buffer (150 mM NaCl, 1% NP40, 50 mM Tris pH 8.0). The samples were diluted in 1X PBS to be measured in a spectrophotometer at 595 nm. The measurement results were reported as arbitrary units of absorbance (AU).

### 4.4. Sphere Assay

The monolayers of MCF-7 and MCF-7Cx46-GFP were harvested and dissociated into single cells by trypsinization. Spheroid formation was performed by seeding the cells at a density of 5 × 10^4^ cells in an ultra-low attachment six-well plate (Corning). Cells were cultured in 2 mL/well of low serum DMEM medium, supplemented with 0.025% of human epidermal growth factor (hEGF), 0.05% insulin, 0.0034% hydrocortisone, and 0.05% heparin. Spheroid formation was monitored each day, and 500 ul of fresh medium was added. The size of each sphere was determined by microscope visualization in a Nikon Eclipse Ti-U inverted microscope (Centro de Fisiología Celular e integrativa, Universidad del Desarrollo, Santiago, Chile). The area of the spheres was measured using NIS-Element AR 4.3 Software (Nikon, Melville, NY, USA).

### 4.5. Migration Assay 

Briefly, 6.0 × 10^5^ cells were plated in 35-mm culture dishes. After 24 h, the cell monolayers were wounded with a 200-μL sterile pipette tip, the cellular debris was washed with PBS, and the medium was replaced with Dulbecco’s modified Eagle’s medium (Life Technologies, Waltham, MA, USA), supplemented with 10% FBS. The wound was photographed at different times (0, 2, 4, 8, and 16 h) with a Nikon Eclipse Ti-U inverted microscope (Centro de Fisiología Celular e integrativa, Universidad del Desarrollo, Santiago, Chile). The area of wound closure was measured using the NIS-Element AR 4.3 Software (Nikon, Melville, NY, USA).

### 4.6. Transwell Invasion Assay

Transwell chambers with 24 wells and an 8 μm polycarbonate membrane (Corning, NY, USA) were used, following the manufacturer’s protocol. The upper chambers were coated with 100 µL of DMEM-diluted Geltrex matrix (Gibco) and incubated at 37 °C for 6 h to allow the gel to solidify. The cultured cells were detached using 0.25% trypsin-EDTA solution and counted in a Neubauer chamber. Then, 1 × 10^5^ cells were seeded into the upper chambers in 200 µL of serum-free media. A total of 500 µL of complete medium was added to the lower chamber as a chemo attractant. After 12 h, the cells remaining in the upper side of the polycarbonate membrane were removed with cotton swabs. Bottom chambers containing invasive cells were washed (twice with PBS), fixed in 100% methanol, and stained with DAPI (5 μm) for 5 min. Ten visual fields of each insert were randomly chosen, and photographed at 4x magnification. The number of cells/fields was quantified using ImageJ software (NIH, Bethesda, MD, USA).

### 4.7. Angiogenesis Assay 

Tube formation in Matrigel (Corning) was used as a functional assay to evaluate the angiogenic properties of MenSCs. Briefly, human umbilical vein endothelial cells (HUVECs) were seeded in 96-well plates precoated with 50 µL of growth factor reduced Matrigel (Corning, Manassas, VA, USA) at a density of 2 × 10^4^ cells/well in the conditioned medium (CM) harvested previously. After 6 h, tube formation was examined by phase contrast, and the images were captured using a Nikon Eclipse Ti-U at 10× magnification. The nodes, junctions, meshes, segments, and branches of the tubes were analyzed using the ImageJ Angiogenesis Analyzer software, version 2.02 (National Institute of Health, Bethesda, MD, USA).

### 4.8. Luminex Assay

SecVEGF was measured using a commercial multiplex fluorescent bead-based immunoassay (R&D Systems, Minneapolis, MN, USA) following the manufacturer’s instructions. The vascular endothelial growth factor A (VEGFA) of the concentrations were determined in CM of MCF-7 and MCF-7Cx46-GFP using a MAGPIX system (ThermoFisher Scientific, Waltham, MA, USA).

### 4.9. Reverse Transcription-Quantitative Polymerase Chain Reaction

Total RNA from the MCF-7 and MCF-7Cx46-GFP was isolated with the Trizol reagent (Life Technologies, Carlsbad, CA, USA). Total RNA concentration was quantified using a Nanodrop (ThermoFisher Scientific, Waltham, MA, USA), following the manufacturer’s instructions. The complementary DNA (cDNA) was generated using SuperScript™ II Reverse Transcriptase (Invitrogen, Carlsbad, CA, USA). The reverse transcription (RT) was performed in 12 µL of reaction: 1 µL of deoxynucleotides (dNTP) (10 mM), 1 µL of random primers (100 ng/µL), 1 µg of total RNA treated with DNAse, and nuclease-free water. The mix was incubated at 65 °C for 5 min. After the incubation, 4 µL of 5X first-strand buffer, 2 µL of Dithiothreitol (DTT) (0.1 M), and 1 µL of RNAseOUT (30 U/µL) were added to the mix. The mix was incubated at 25 °C for 2 min, and 1 µL of SuperScript™ II Reverse Transcriptase (200 U) was added. The complete reaction was incubated at 25 °C for 10 min, 42 °C for 50 min, and inactivated at 70 °C for 15 min. Quantitative real-time PCR (qPCR) was performed using the Brilliant II QPCR Master Mix (Agilent Technologies, Santa Clara, CA, USA) mixing 2.5 µL of diluted cDNA, 200 nmoles of each primer, 5 µL of Brilliant II QPCR Master Mix, and nuclease-free water in a final volume of 10 µL. The reaction was incubated 10 min at 95 °C, 40 cycles of 20 s at 95 °C, 20 s at 60–62 °C, 20 s at 72 °C, and finally 10 s at 95 °C, 5 s to 25 °C, 1 s at 55 °C, and 1 s at 95 °C in the Stratagene Mx3000P system (Agilent Technologies, Santa Clara, CA, USA). The primers used to analyze the expression of the epithelial-to-mesenchymal transition genes were: N-Cad: forward 5′-CGGGAGAAATTGCAGGAGGA-3′, reverse 5′-GGCAAGTTGATTGGAGGG ATG-3′, Vimentin: forward 5′-CGGGAGAAATTGCAGGAG GA-3′, reverse 5′-AAGGTCAAGACGTGCCAGAG-3′, SNAIL: forward 5′-CTTCCA GCAGCCCTACGAC-3′, reverse 5′-GACAGAGTCCCAGATGAGCA-3′, and Zeb1: forward 5′-AAGTGGCGGTAGATGGTAATG-3′, reverse 5′-AGGAAGACTGATGGCTGAAATAA-3′, and the primers used to analyzed the expression of the stemness genes were: Oct4: forward 5′-AGGTATTCAGCCAAACGACCA-3′, reverse 5′-TCGATACTGGTTCGCTTTCTC-3′, Nanog: forward 5′-CATGAGTGTGGATCCAGCTTG-3′, reverse 5′-CCTGAA TAAGCAGATCCATGG-3′, and Sox2: forward 5′-AGCTACAGCATGATGCAGGA-3′, reverse 5′-GAGTAGGACATGCTGTAGGT-3′, Glyceraldehyde 3-phosphate dehydrogenase (GAPDH) was used as a housekeeping gene: forward 5′-GGAAGATGGTGATGGGATTTC-3′, reverse 5′-GAAGGTGAAGGTCGGAGTCAA-3′. All the primers were synthesized by IDT DNA technologies (Coralville, IA, USA). Transcript levels were quantified using the comparative CT method.

### 4.10. Statistical Analysis

All results are expressed as the mean ± standard error of the mean (SEM). Data were processed using GraphPad software, Inc. (http://www.graphpad.com, accessed on 5 November 2021) and analyzed for statistical significance using the student’s *t*-test; a *p*-value < 0.05 was considered significant. 

## 5. Conclusions

We show here that Cx46 expression enhances the CSC, EMT, and angiogenic characteristics on MCF-7 breast cancer cells. It increases the transcription factors associated with CSC and EMT. Additionally, Cx46-expressing cells release more VEGF to the extracellular milieu, where it induces angiogenesis in Huvec cells. Our data indicate an important role of Cx46 as a driver of cancer aggressiveness.

## Figures and Tables

**Figure 1 ijms-22-12604-f001:**
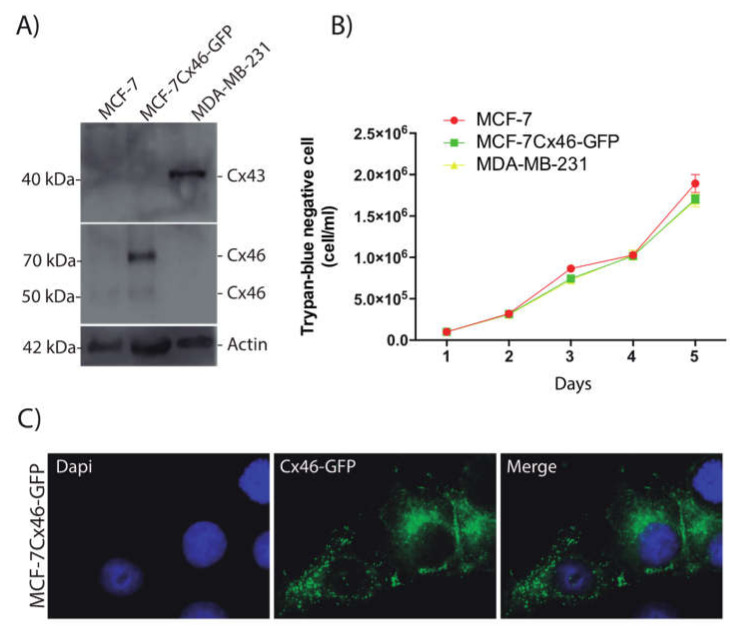
Connexin 46-GFP overexpressed in MCF-7 cells does not modify the rate of cell division. MCF-7 breast cancer cells were transfected with the human connexin Cx-46 gene inserted in a pcmV6-AC-GFP plasmid and selected by neomycin resistance. (**A**) Representative images of three independent Western blots of MCF-7, MCF-7Cx46-GFP, and MDAMB-231 cell lysates; immunoblots were incubated with anti Cx46, and anti-beta Actin (Santa Cruz Biotechnology; 1:2500). Protein bands were detected using Immobilon Forte Western horseradish protein (HRP) substrate and visualized with a LI-COR CDigit System. (**B**) A trypan blue dye exclusion test determined the number of viable cells. MCF-7, MCF7Cx46-GFP, and MDAMB-231 were seeded, and the trypan blue negative cells were counted in a Neubauer chamber. The cells were counted every day until the fifth day. (**C**) Representative fluorescence images of MCF-7Cx46-GFP cell nuclei were visualized with Dapi (left), and Cx46 was visualized with the GFP tag in the C-terminal portion of Cx46 (middle). Images were obtained using a Nikon Eclipse Ti-U inverted microscope. Data are presented as the mean of three independent experiments +/− SEM.

**Figure 2 ijms-22-12604-f002:**
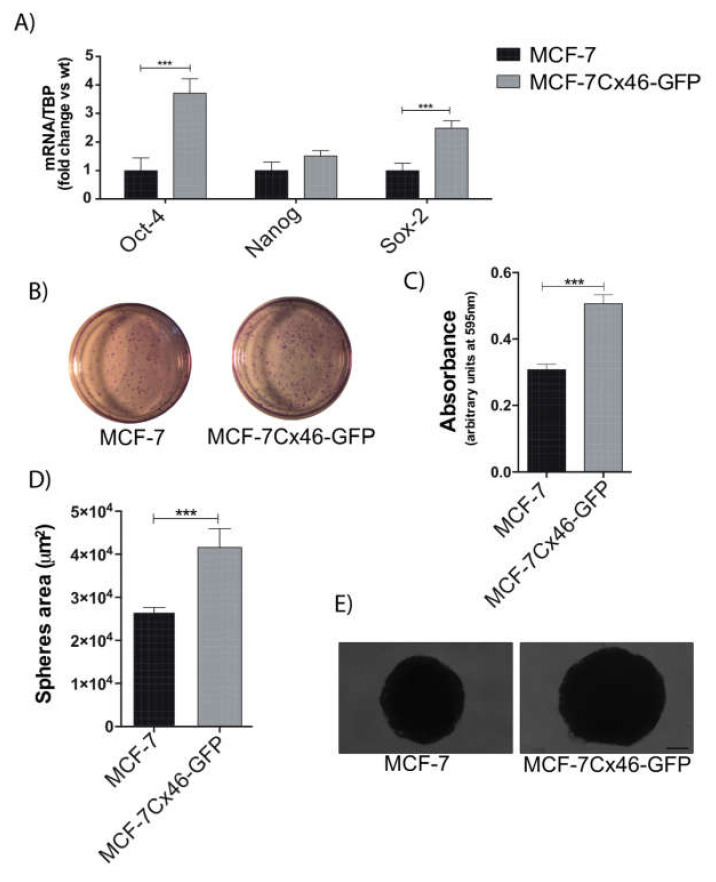
Cx46 increases the expression of CSC-related genes and CSC stemness. (**A**) Cancer stem cell-related genes were evaluated by measuring the mRNA levels of Oct4, Nanog, and Sox2 genes by qRT-PCR in MCF-7 and MCF-7Cx46-GFP breast cancer cells; data represent the mean of three independent experiments +/− SEM. (**B**) Clonogenic potential of MCF-7 and MCF-7Cx46-GFP breast cancer cells. Representative images of clonogenic assay were fixed and stained with crystal violet solution. (**C**) Cells were dissolved with lysis buffer and absorbance was measured at 595 nm (arbitrary units at 595 nm). The results represent the means of three independent experiments +/− SEM. (**D**) The graph represents the comparison of sphere diameters between MCF-7 and MCF-7Cx46-GFP before 3 weeks of culture. (**E**) Representative image of 3-week sphere morphology of MCF-7 and MCF-7Cx46-GFP cells; scale bar 100 μm. Data are presented as the mean of three independent experiments +/− SEM. *** *p* < 0.001.

**Figure 3 ijms-22-12604-f003:**
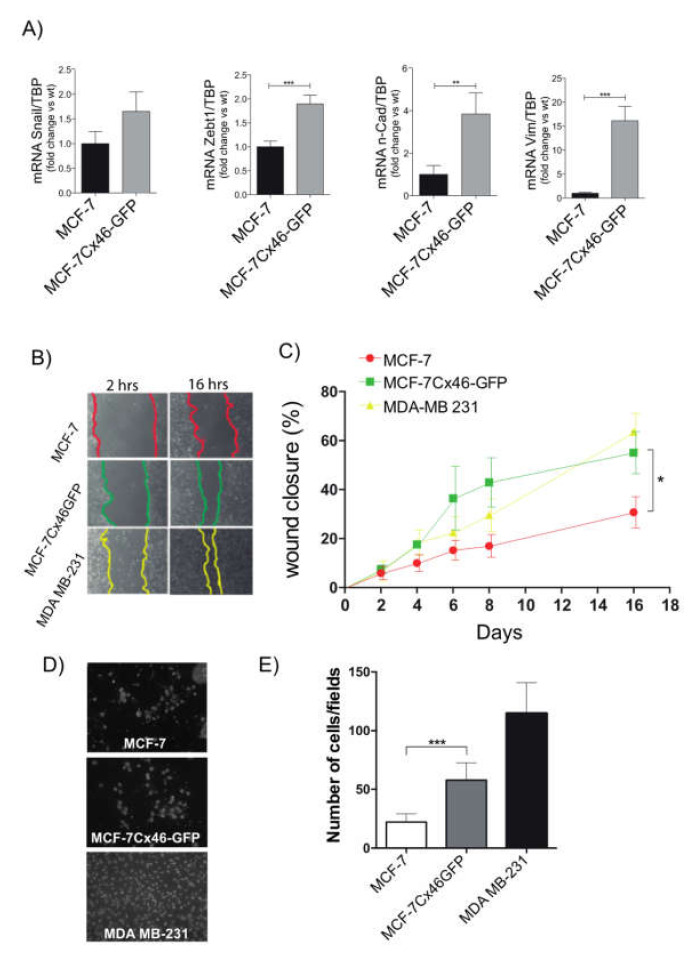
Cx46-GFP increases the expression of EMT-related genes and EMT functional characteristics. (**A**) The EMT genes Snail, Zebt1, N-Caderin and Vimentin of MCF-7 and MCF-7Cx46-GFP were determined by qRT-PCR. The graph represents the means of three independent experiments. (**B**) In vitro scratch assays determined differences in breast cancer cell migration. The image in (**B**) was recorded every 2 h and photos were obtained at the indicated time points using a 10× objective on a Nikon inverted microscope, recorded and measured using the NIS-Element software. The figure shows images at 2 and 16 h for each cell line, and the color line shows the gap area in the three different cell lines evaluated. (**C**) Graphical representations of percentage of wound closure area of the three different experiments represented in B. The closing percentage was calculated at different times until the 16 h period was complete. Data represent the means of three independent experiments +/− SEM. * denotes *p* < 0.05, ** *p* < 0.01 and *** *p* < 0.001. (**D**) Representative images of fields in a Transwell migration assay of MCF-7 (up), MCF-7Cx46-GFP (center), and MDAMB-231 (bottom); cells that crossed the gel matrix were fixed and stained with Dapi. (**E**) Ten different fields were photographed, and the number of cells was counted and graphed. Data are presented as the means of three independent experiments +/− SEM.

**Figure 4 ijms-22-12604-f004:**
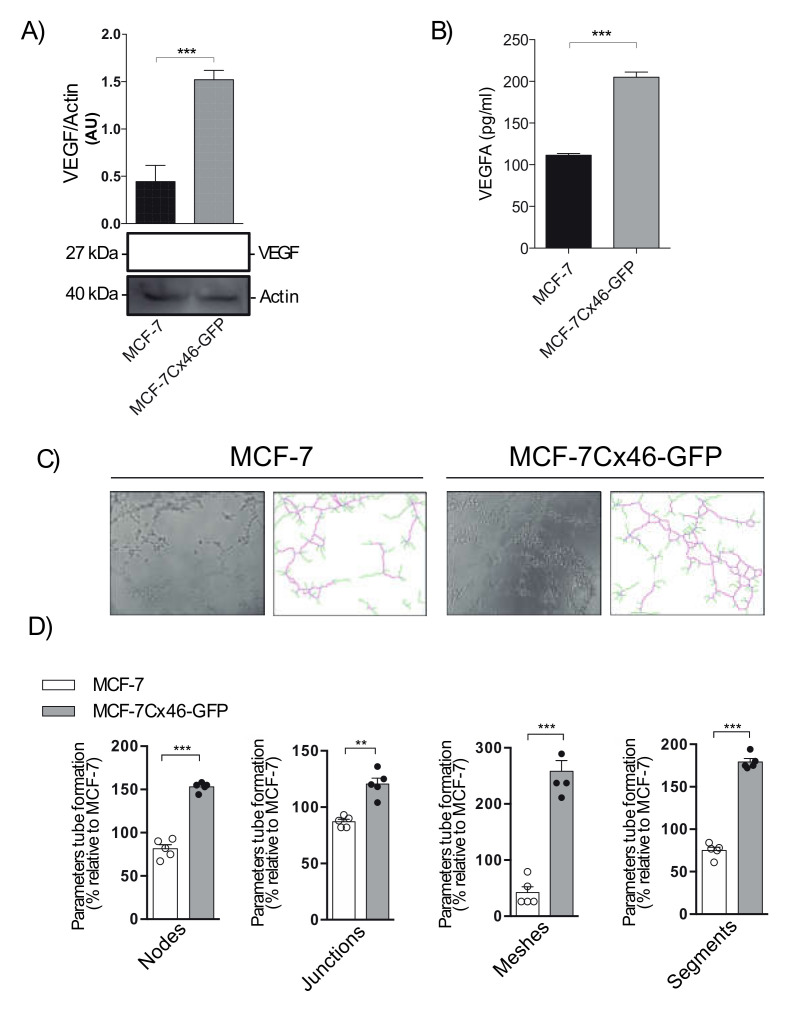
Cx46 increases the expression of angiogenic factors. (**A**) Representative Western blot of VEGF expression in MCF-7 and MCF-7Cx46-GFP and densitometry quantification of three independent Western blots of VEGF protein expression. (**B**) Graphical representation of VEGF release in conditioned supernatants media evaluated using a Luminex MAGPIX system. (**C**) Representative images of tubule structure formation in HUVEC cell incubated with conditioned medium from MCF-7 (left) and MCF-7Cx46-GFP (right) cells. (**D**) Representative graph quantification of different tube formation parameters (junctions, nodes, meshes, and branches) in HUVEC cells incubated with conditioned medium from MCF-7 and MCF-7Cx46-GFP. After 16 h, the tube formation was examined by phase contrast and the images were captured using an Olympus U-RFL-T camera. Junctions, nodes, meshes, and branches were analyzed using the ImageJ Angiogenesis Analyzer software. Data are presented as the means of three independent experiments +/− SEM. ** *p* < 0.01 and *** *p* < 0.001.

## Data Availability

Data will be available if requested.

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
