# Peer review of "Connexin46 Expression Enhances Cancer Stem Cell and Epithelial-to-Mesenchymal Transition Characteristics of Human Breast Cancer MCF-7 Cells"

_ijms, 2021, doi:10.3390/ijms222212604_

Round 1

Reviewer 1 Report

Some concerns and questions are listed below to be addressed;

  1. Please show the chemo-sensitivity change between wild type and Cx46-overexpressed MCF7 cells
  2. Please show protein data at Fig.2A and Fig3A.
  3. Tumorsphere assay could not evaluate the sphere size, which represent the initial involvement of non-stem cells.  Please show the sphere number differences and pictures (Fig.2 D, E).
  4. Please show the more gene signature contrasting between wild type and Cx46-overexpressed MCF7 cells.
  5. Please show the mechanistic background why Cx46 overexpression promotes EMT and cancer stemness.

Author Response

I greatly appreciate the comments made on my proposal, the answers are in the attached file

Reviewer 2 Report

The paper aims to examine the mechanism by which Cx46 could promote stem cells in in breast cancer. For that, the authors compare different aspects of MCF-7 cells overexpressing Cx46-EGFP and original MCF-7 cells.

Major concerns

  1. The original cells are not exactly reference cells it would be appropriate to use cells over expressing GFP
  2. It is known that GFP affect different properties of connexin if GFP is directly linked to connexins.

Therefore the authors would be more convincing if they could reproduce some key results for example the expression of stem cell markers, release of VEGF, angiogenesis, etc using cells expressing Cx46 separated from GFP using cells expressing GFP as reference

  1. In different figures is stated: “Data are presented as the mean +/- SEM” for which “n”?
  2. Cx have different functions, which function is involved in the different aspects presented in the paper?

Minor concerns

  1. Do the authors mean really “cystidium” or they mean “syncytium”
  2. Figure 4 the legend for A is the legend for b and vice versa

Author Response

(The authors gave the same response as above.)

Round 2

Reviewer 2 Report

My major is the linked the connexin and GFP. I do not expect the authors to perform all the experiments  using cells in which GFP and the connexin are separated. However the authors could show that the separation Cx and GFP does affect some important results e.g wound healing or tubule formation. 

Round 3

Reviewer 2 Report

The authors have considered my concerns